# *Paeniclostridium sordellii* uterine infection is dependent on the estrous cycle

**Sarah C. Bernard**[1], **M. Kay Washington**[1], **D. Borden Lacy**[1,2]*

**1** Department of Pathology, Microbiology, and Immunology, Vanderbilt University Medical Center, Nashville, Tennessee, United States of America, **2** Veterans Affairs Tennessee Valley Healthcare System, Nashville, Tennessee, United States of America

* borden.lacy@vanderbilt.edu

## Abstract

Human infections caused by the toxin-producing, anaerobic and spore-forming bacterium *Paeniclostridium sordellii* are associated with a treatment-refractory toxic shock syndrome (TSS). Reproductive-age women are at increased risk for *P. sordellii* infection (PSI) because this organism can cause intrauterine infection following childbirth, stillbirth, or abortion. PSI-induced TSS in this setting is nearly 100% fatal, and there are no effective treatments. TcsL, or lethal toxin, is the primary virulence factor in PSI and shares 70% sequence identity with *Clostridioides difficile* toxin B (TcdB). We therefore reasoned that a neutralizing monoclonal antibody (mAB) against TcdB might also provide protection against TcsL and PSI. We characterized two anti-TcdB mABs: PA41, which binds and prevents translocation of the TcdB glucosyltransferase domain into the cell, and CDB1, a biosimilar of bezlotoxumab, which prevents TcdB binding to a cell surface receptor. Both mABs could neutralize the cytotoxic activity of recombinant TcsL on Vero cells. To determine the efficacy of PA41 and CDB1 *in vivo*, we developed a transcervical inoculation method for modeling uterine PSI in mice. In the process, we discovered that the stage of the mouse reproductive cycle was a key variable in establishing symptoms of disease. By synchronizing the mice in diestrus with progesterone prior to transcervical inoculation with TcsL or vegetative *P. sordellii*, we observed highly reproducible intoxication and infection dynamics. PA41 showed efficacy in protecting against toxin in our transcervical *in vivo* model, but CDB1 did not. Furthermore, PA41 could provide protection following *P. sordellii* bacterial and spore infections, suggesting a path for further optimization and clinical translation in the effort to advance treatment options for PSI infection.

## Author summary

*P. sordellii* infection (PSI) in humans is rare but typically fatal. It is most frequently observed as a uterine infection in postpartum women following childbirth, stillbirth, or abortion. Once identified, antibiotics can be used to eradicate the bacteria, but these are not effective at neutralizing the secreted TcsL lethal toxin that can cause a treatment-refractory toxic shock syndrome. A neutralizing antibody against TcsL would address this

**Data Availability Statement:** All relevant data are within the manuscript and its Supporting Information files.

**Funding:** We gratefully acknowledge funding support from NIH T32 AI007281 (SCB) and

Vanderbilt University Medical Center. Research in the Lacy lab is supported by the NIH (AI095755) (DBL) and the Department of Veterans Affairs (BX002943) (DBL). The Vanderbilt University Medical Center's Digestive Disease Research Center is supported by NIH grant P30DK058404 (MKW). The funders had no role in the study design, data collection and analysis, decision to publish, or preparation of the manuscript.

**Competing interests:** The authors have declared that no competing interests exist.

problem in concept but has never been directly tested. TcsL shares high sequence identity with *C. difficile* TcdB, the primary virulence factor in *C. difficile* infection (CDI). We characterized two *C. difficile* TcdB antibodies, CDB1, a biosimilar to a clinically available drug, and PA41, in the neutralization capabilities of TcsL. In the process, we established a non-surgical, uterine infection model and made the discovery that disease symptoms varied with the reproductive cycle of the animals. This opens the door for new research questions at the interface of bacterial spore and toxin biology with reproductive health. While CDB1 did not provide protection against PSI in our animal model, PA41 did show protection. If developed for CDI, this antibody could have an added therapeutic utility in the life-threatening instances of human PSI.

## Introduction

Human infections caused by the toxin-producing, anaerobic and spore-forming bacterium *Paeniclostridium sordellii* are associated with a treatment-refractory toxic shock syndrome (TSS) and are typically lethal [1]. Reproductive-age women are at increased risk for *P. sordellii* infection (PSI) because this organism can cause intrauterine infection following childbirth or abortion [1]. Clinical indications of disease include a marked leukemoid reaction, i.e., a vast increase in white blood cells, increased vascular permeability, hemoconcentration, and, in most cases, the absence of a fever [1]. When women present with PSI-TSS, very little is known on how to treat the patient [1]. In most cases, a hysterectomy is performed along with definitive antibiotic therapy. However, even if antibiotics are successful in killing the bacteria, there are bacterial toxins that can continue circulating the body to cause disease.

*P. sordellii* secretes two cytotoxins that are similar in structure and function to toxins generated by the pathogen *Clostridioides difficile*: lethal toxin (TcsL), similar to *C. difficile* TcdB, sharing 76% sequence identity, and hemorrhagic toxin (TcsH), similar to *C. difficile* TcdA, sharing 78% sequence identity. Both TcsL and TcsH, like the *C. difficile* toxins, are glucosyltransferases that inactivate host GTPases. Some TcsL-positive isolates lack the gene encoding TcsH and are rapidly lethal in an animal model, indicating that TcsH is not essential for virulence [2]. Genetically derived TcsL-mutant strains were nonlethal in a mouse infection model giving evidence that TcsL is an essential virulence factor responsible for disease in PSI [3]. Neutralizing the cytopathic effect of TcsL might protect humans against toxic shock caused by TcsL-expressing *P. sordellii*.

Two anti-TcdB monoclonal antibodies, PA41 and CDB1, have been characterized and shown to neutralize TcdB in cell culture and animal models [4–6]. PA41 binds the glucosyltransferase domain (GTD) of TcdB, inhibiting the delivery of the enzymatic cargo into the host cell [4]. CDB1, a mAB whose Fab sequence is identical to that of Bezlotoxumab, neutralizes TcdB by blocking binding of TcdB to mammalian cells [6]. Given the high levels of sequence identity between TcsL and TcdB, we wondered if these anti-TcdB antibodies would also neutralize TcsL, and if so, if they would provide protection in an animal model of infection. We were particularly interested in the potential of CDB1, as the mAB Bezlotoxumab is an FDA approved therapeutic for the prevention of CDI recurrence [7]. To have a clinically available mAB that also targets TcsL, the key virulence factor in PSI, could represent a significant tool in the limited therapeutic arsenal when faced with human PSI.

Developing effective interventions against PSI (and TSS) is stymied by a lack of animal models and an incomplete understanding of how *P. sordellii* induces disease. Some investigators have used intraperitoneal injection of toxin [8,9], but this model is not optimal in terms of

physiological relevance. To increase relevancy, a uterine mouse model was established to study *PSI*-associated TSS [10]. This intrauterine infection involves survival surgery to allow for ligation at the cervical junction and direct introduction of bacteria into the uterine lumen [10]. This model, however, provides additional pain and stress to the animals and increases the risk of infecting the blood stream directly. To address this problem, we developed an innovative mouse model system in which to study PSI using a transcervical (TC) inoculation method. Such a model allows for a non-surgical transfer of inoculum through the vaginal orifice, past the cervix, and directly into the uterus. This method eliminates the need for intensive survival surgery and produces a disease that more closely represents the nature of PSI in postnatal and post-abortive women.

## Results

### Neutralization of recombinant TcsL by monoclonal antibodies, PA41 and CDB1, *in vitro*

To assess the effects of anti-TcdB antibodies PA41 and CDB1 on TcsL *in vitro*, cell neutralization assays were performed. First, Vero cells were treated with serial dilutions of TcsL in the presence and absence of 100 nM concentrations of the mABs for 72 hours and then assayed for ATP levels as an indicator of viability (Fig 1A). In panel B of Fig 1, we display the same data as a bar graph, indicating the concentrations of TcsL where we see a statistically significant difference. At 100 fM TcsL, PA41, but not CDB1, provided a statistically significant improvement in viability, although TcsL is not very toxic at this concentration. The cells, however, were very sensitive to 1 pM TcsL, and PA41 was able to completely neutralize the cytotoxic activity of the toxin. CDB1, also, was able to show statistically significant neutralization at this TcsL concentration, though to a lesser extent than PA41. Vero cells were not viable following intoxication with 10 pM and 100 pM TcsL alone. In the presence of PA41, cells were completely or partially viable when intoxicated with 10 pM or 100 pM TcsL, respectively. CDB1 was not able to protect at 10 pM or 100pM TcsL. Altogether, both antibodies neutralized the cytotoxicity of TcsL on Vero cells, though PA41 appeared more effective than CDB1 in all conditions. We also performed a dose titration of both antibodies at a cytotoxic dose of TcsL (1 pM) to get an understanding of their relative potencies (Fig 1C and 1D). Strong neutralization of TcsL was observed with 100 pM PA41. Following a titration of CDB1, we report a sharp reduction of antibody potency below 100 nM.

### PA41 and CDB1 neutralization of TcsL, *in vivo*, following intraperitoneal injection

To assess the effects of PA41 and CDB1 on TcsL *in vivo*, it was first necessary to determine the lowest lethal intraperitoneal (IP) dose of TcsL. Female, 9-12-week-old, C57BL/6J mice were IP injected with 1 ng and 2.5 ng TcsL in 100 uL PBS (Fig 2A). In this study, 2.5 ng TcsL was the lowest lethal dose administered, with all animals succumbing to intoxication prior to 24 hours post administration. Consistent with previous findings [9], all mice intoxicated with this amount and higher of TcsL had a buildup of fluid in the thoracic and peritoneal cavities (S1 Fig). Mice intoxicated with 1 ng TcsL survived the study and showed no signs of disease.

Next, we wanted to perform a survival study with 2.5 ng TcsL IP alone or in the presence of PA41 or CDB1 (Fig 2B). Following our *in vitro* assays, we determined a 10000-fold excess (0.75mg/kg) of antibody to lethal TcsL (2.5ng) would be a reasonable dose to use in our *in vivo* IP intoxications. We found that PA41 was able to effectively neutralize TcsL, and all the animals survived the study with no signs of disease. CDB1 was not able to completely neutralize

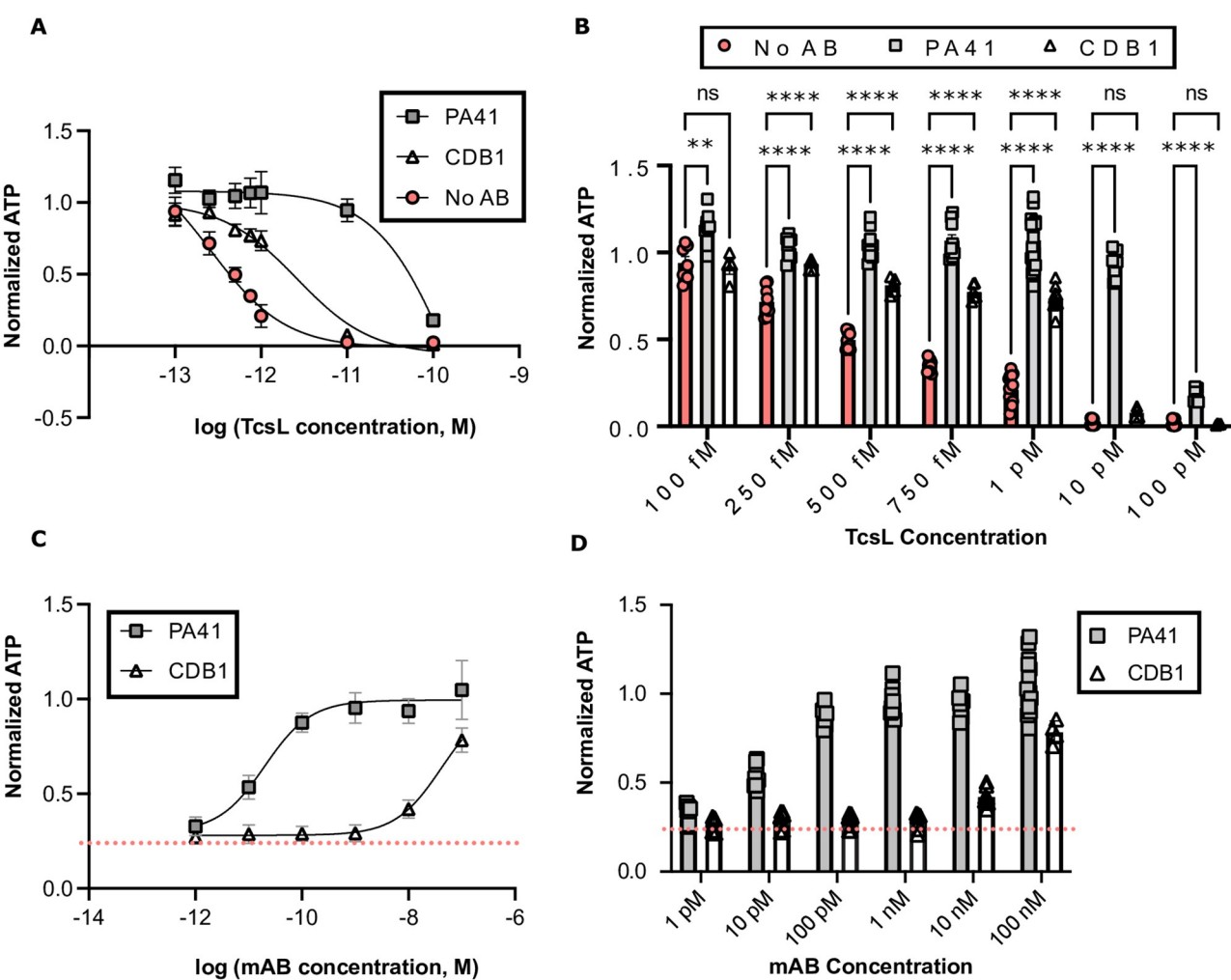

**Fig 1. Neutralization of TcsL cytotoxicity by *C. difficile* monoclonal antibodies, PA41 and CDB1, *in vitro*.** (A-B) Vero cells were treated with serial dilutions of TcsL alone or in the presence of 100 nM PA41 or CDB1. ATP was measured as a readout of viability and normalized to signal from untreated cells. Dunnett's test for multiple comparisons was used with statistical significance set at a p value of <0.05, where **** represents p <0.0001. (C-D) Vero cells were treated with a serial dilution of PA41 or CDB1 in the presence of 1 pM TcsL (IC$_{50}$ = 20 pM and ~41 nM, respectively). The baseline for 1 pM TcsL cytotoxicity is indicated by the dotted line.

TcsL when given at 0.75 mg/kg, resulting in a survival of 66%. This result led us to increase the antibody dose ten-fold to 7.5 mg/kg to be used for all in vivo studies moving forward to give both antibodies the best opportunity for neutralization. To better understand how long the antibody circulates in the bloodstream following a single administration of PA41, we performed a three-day study where on Day 0, three animals were administered 7.5 mg/kg PA41. On days 1, 2, and 3, an animal was euthanized and the whole blood was collected. From western blot analysis of the serum using an anti-human Fab antibody, we could consistently observe PA41 in serum each day for up to three days without showing signs of depletion (S2 Fig).

From these *in vivo* studies, both *C. difficile* monoclonal antibodies were able to neutralize 2.5 ng TcsL following IP injection, with PA41 appearing more effective than CDB1. In addition, antibody administration protected mice from TcsL-induced pleural effusion.

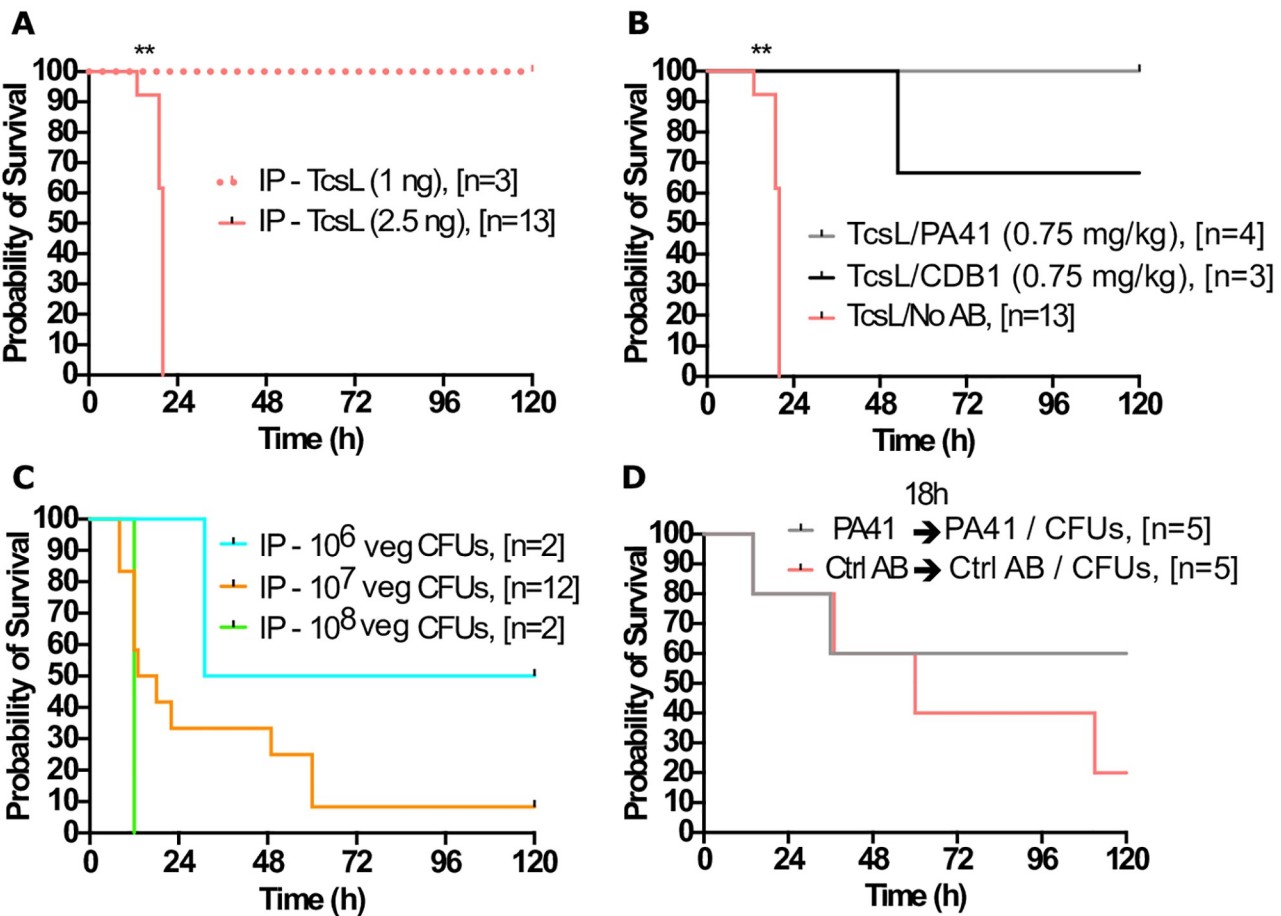

**Fig 2. PA41 and CDB1 neutralization of rTcsL and P. sordellii vegetative bacteria, *in vivo*, following intraperitoneal injection. (A)** Mouse survival curve following IP injection of 1ng and 2.5ng TcsL. **(B)** Mouse survival curve following IP intoxication of 2.5ng TcsL alone or in the presence of PA41 (0.75 mg/kg) or CDB1 (0.75 mg/kg). **(C)** Mouse survival curve following IP infections of $10^6$, $10^7$, $10^8$ CFUs of vegetative *P. sordellii* strain ATCC 9714. **(D)** Mouse survival curve following 7.5 mg/kg PA41 or PA50 (control antibody) administered 18h prior to co-IP instillation of 7.5 mg/kg antibody and $1 \times 10^7$ CFUs of ATCC 9714 vegetative bacteria. Log-rank (Mantel-Cox) multiple comparison test was used with statistical significance set at a p value of $<0.05$.

### *In vivo* neutralization of *P. sordellii* vegetative bacteria following intraperitoneal injection

Our next steps were to infect mice with *P. sordellii* vegetative bacteria and assess whether PA41 and CDB1 offered protection. We chose to infect with a highly virulent *P. sordellii* reference strain, ATCC 9714, that lacks the gene for TcsH. Intraperitoneal injections of $10^6$, $10^7$, $10^8$ CFUs vegetative bacteria were administered to determine the number of bacteria to use in neutralization studies (Fig 2C). We found that injection of $10^7$ CFUs resulted in a survival curve that was penetrant with 11 of the 12 infected mice dying over the course of 60 hours. Injection of $10^8$ CFUs resulted in death by 12hr, a time we predicted to be too short to allow for mAB neutralization. A lower bacterial count of $10^6$ CFUs, resulted in only 50% survival, but with only two animals tested.

Next, we tested antibody neutralization of $10^7$ CFUs *P. sordellii* vegetative bacteria. Since PA41 showed the most efficacy in TcsL neutralization *in vitro* and *in vivo*, these studies were done with PA41 (7.5 mg/kg). It is plausible that by using vegetative bacteria, TcsL is already being produced and may overwhelm the antibody when administered at the same time. To

reduce this possibility, antibody was administered by IP injection 18 hr prior to IP injection of both vegetative bacteria and a second dose of antibody. PA50, a monoclonal antibody against *C. difficile* TcdA, was used as a negative control [11]. In this experiment, the PA41-treated mice had a marginally higher survival rate when compared to control-treated mice (Fig 2D). This result, though, was not found to be statistically significant.

## Transcervical instillation of recombinant TcsL or vegetative *P. sordellii* to study uterine PSI

Having observed some efficacy with the antibodies using the IP models, we next wanted to test their effectiveness in a more physiologically relevant animal model. We developed an innovative mouse model system in which to study PSI using a transcervical (TC) inoculation method. The model allows for a non-surgical transfer of inoculum through the vaginal orifice, past the cervix, and directly into the uterus (Fig 3A). For TC instillation, a speculum was inserted into the vaginal cavity to allow for dilation and passage of a gel loading pipette tip through the cervix and transfer of inoculum directly into the lumen of the uterine horn. This method enabled the simple instillation of vegetative bacteria, spores, or recombinant protein into the uterus. Following instillation, a cotton plug applicator was inserted into the vagina, and the cotton plug was expelled from the applicator and into the vaginal cavity using a blunt needle. This cotton plug was used as an absorptive material to keep inoculum in the reproductive tract and to minimize any leakage into the environment.

While 2.5 ng TcsL IP injections resulted in death before 24 hours post-intoxication (Fig 2A), the TC instillation of 5, 25, or 50 ng TcsL did not result in any signs of disease or death (Fig 3B). This suggested that TcsL alone is not cytotoxic to the epithelium of the reproductive tract, but perhaps requires assistance from other *P. sordellii* virulence factors. To test this hypothesis, TC inoculations of *P. sordellii* strain ATCC 9714 vegetative bacteria were performed. However, instillations of either $10^7$ or $10^8$ CFUs resulted in minimal death/signs of infection as compared to IP infection (Fig 3C).

## The murine reproductive cycle determines the pathogenic outcome of *P. sordellii* uterine intoxications and infections

We next tested whether manipulation of the host hormonal environment influences the murine susceptibility to TcsL intoxication and *P. sordellii* infection. For estrous cycle synchronization, medroxyprogesterone acetate was administered subcutaneously five days prior to intoxication/infection to prolong diestrus, and beta-estradiol was administered subcutaneously two days prior to intoxication/infection to prolong estrus. Immediately prior to instillation, we confirmed via vaginal lavage analysis that the mice were in the expected stage of the reproductive cycle. Animals were weighed and monitored daily for six to eight days (Fig 4A). To begin, animals in diestrus or estrus were transcervically instilled with 50 ng TcsL. All animals in diestrus succumbed to intoxication by 24h (Fig 4B). Conversely, all animal in estrus survived the study with no signs of disease or sickness. Diestrus animals were then subjected to 5, 10, 20, 50 and 500 ng TcsL, and their resulting survival curves revealed increasing severity with each increase in dose. (Fig 4C). Next, animals in diestrus or estrus were transcervically inoculated with $10^7$ CFUs *P. sordellii* 9714 vegetative bacteria. A statistically significant difference was found between animal in diestrus compared to estrus, with animals in diestrus having a more adverse outcome to infection compared to animals in estrus (Fig 4D). A bacterial titration of vegetative bacteria was administered TC to animals in diestrus and the resulting survival curves revealed $10^7, 10^6$, and $10^5$ CFUs to be similar in terms of severity [~15–30% survival] and $10^4$ CFUs to be less severe [80% survival], followed by $10^2$ CFUs which did not cause any

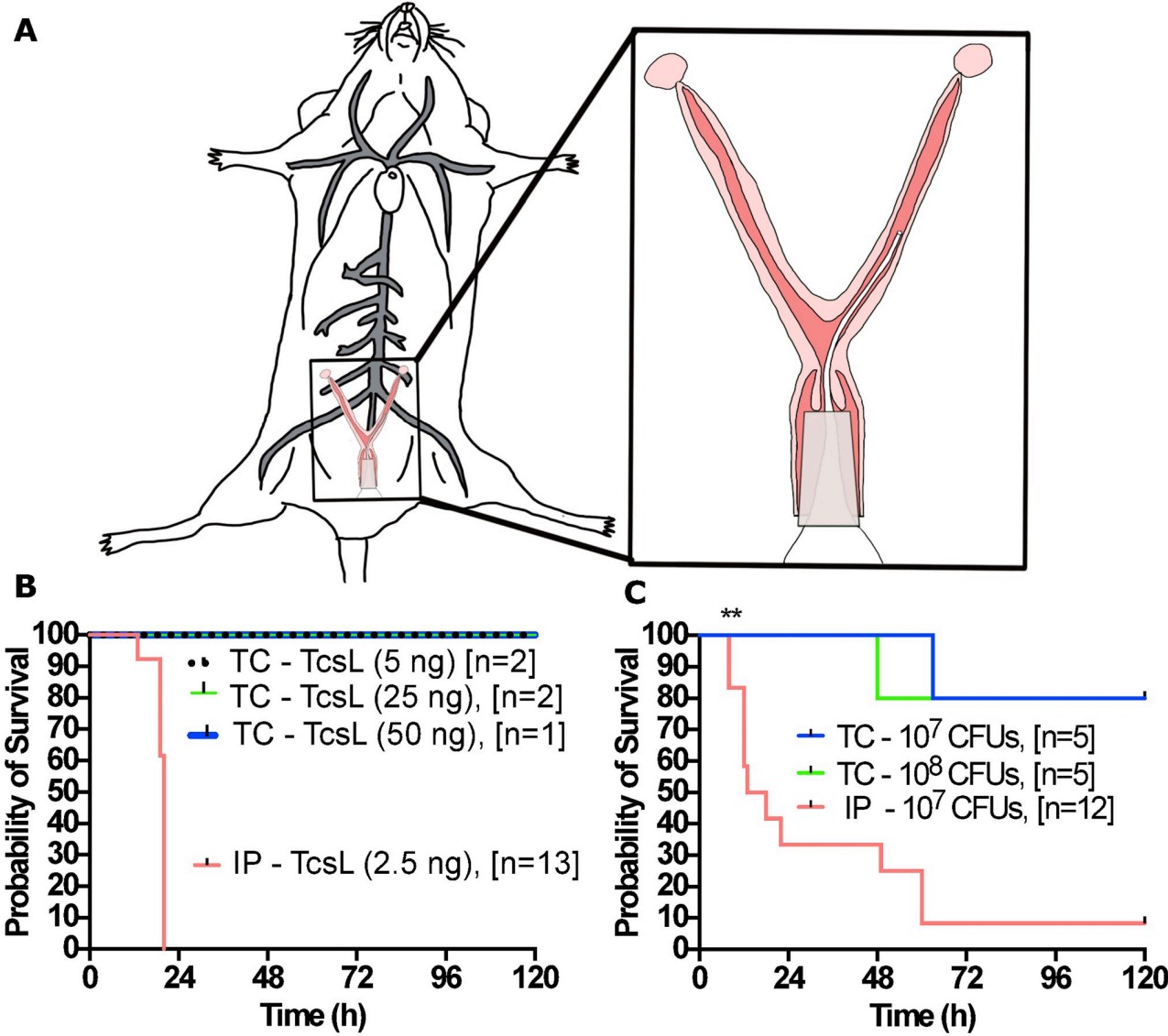

**Fig 3. Transcervical instillation method of recombinant TcsL or vegetative *P. sordellii*. (A)** Schematic depicting the murine transcervical instillation method. **(B)** Transcervical intoxication of 5, 25, 50ng rTcsL and intraperitoneal intoxication of 2.5ng rTcsL. **(C)** Transcervical infection of $10^7$ and $10^8$ CFUs and intraperitoneal infection of $10^7$ CFUs of vegetative ATCC 9714 *P. sordellii*.

detectable signs of sickness in the animals (Fig 4E). From these experiments, we conclude that the mouse reproductive cycle can influence the pathogenic outcome of TcsL intoxications and uterine *P. sordellii* infections.

## PA41 and CDB1 neutralization studies of TcsL, *in vivo*, following transcervical instillation of animals in diestrus

We next sought to determine the efficacy of PA41 and CDB1 in neutralization of TcsL intoxication in our hormone-inducing transcervical instillation model. All animals were administered medroxyprogesterone acetate five days prior to instillation to induce diestrus. For antibody administration, since we know TcsL is rapidly lethal (Fig 4C), we wanted to have a higher amount of PA41 in the bloodstream prior to intoxication. We tested a single IP

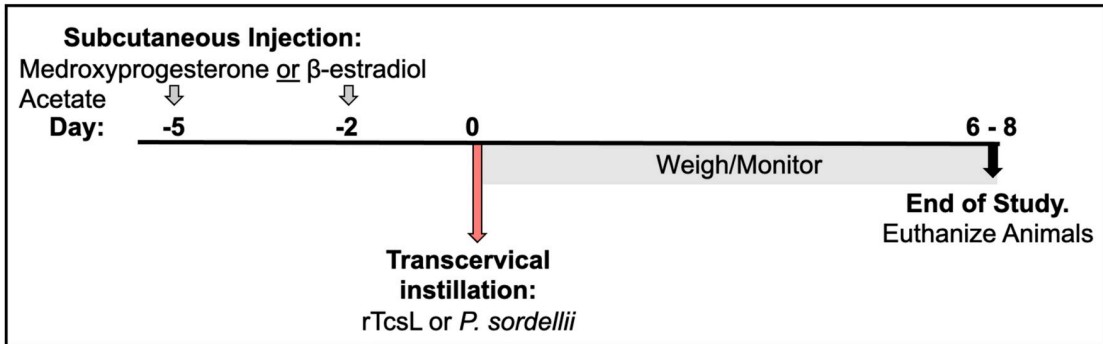

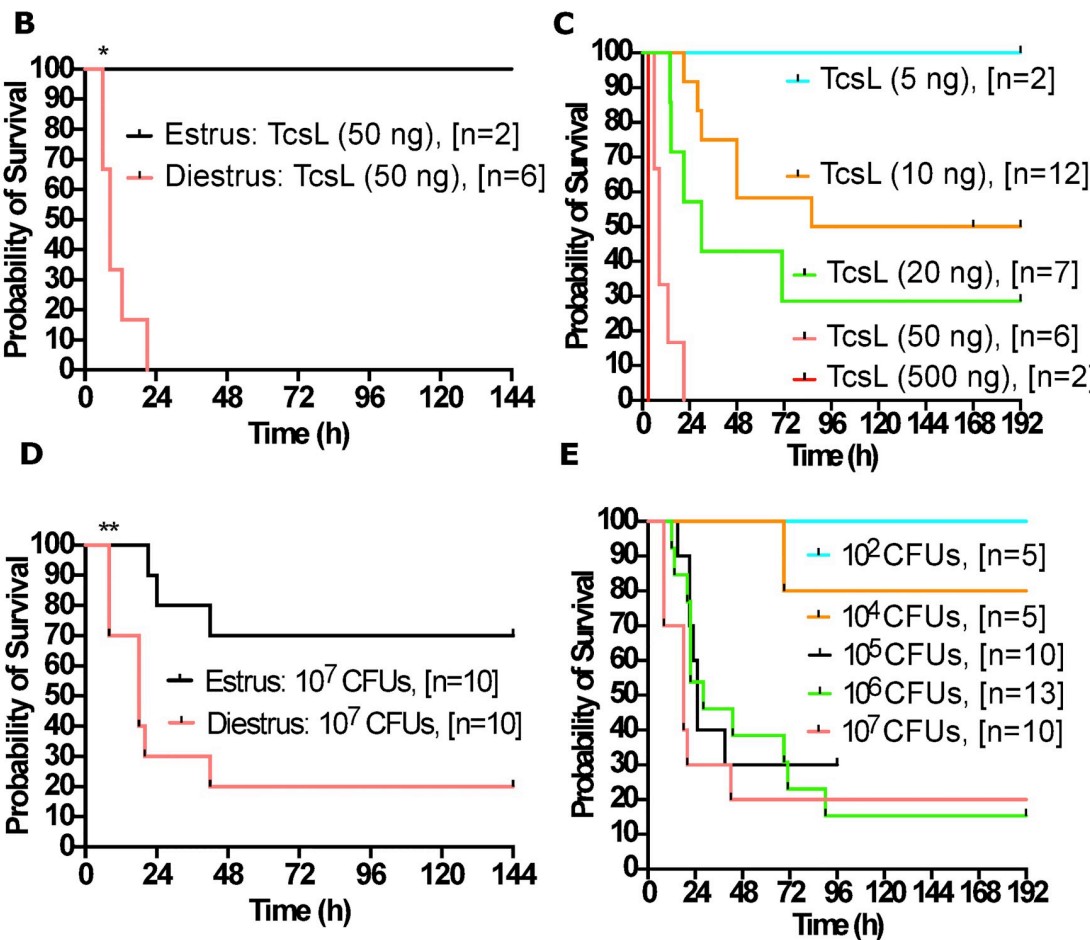

**Fig 4. Development of hormonal transcervical instillation method of recombinant TcsL or vegetative *P. sordellii*. (A)** Timeline of hormonal synchronization of murine estrous cycles following subcutaneous administration of medroxyprogesterone acetate (Day -5) to induce diestrus or beta-estradiol (Day -2) to induce estrus. Transcervical Instillation of rTcsL or P. sordellii was performed on Day 0 and animals were weighed/monitored for six to eight days. **(B)** Survival curve of mice subcutaneously injected with medroxyprogesterone acetate or beta-estradiol followed by transcervical intoxication with 50ng TcsL. **(C)** Survival curve of mice subcutaneously injected with medroxyprogesterone acetate followed by transcervical intoxication with 5, 10, 20, 50, 500ng TcsL. **(D)** Survival curve of mice subcutaneously injected with medroxyprogesterone acetate or beta-estradiol followed by transcervical infection with $10^7$ CFUs vegetative *P. sordellii* ATCC9714. **(E)** Survival curve of mice subcutaneously injected with medroxyprogesterone acetate followed by transcervical infection with $10^2$, $10^4$, $10^5$, $10^6$, $10^7$ CFUs vegetative *P. sordellii* ATCC9714. Log-rank (Mantel-Cox) multiple comparison test was used with statistical significance set at a p value of <0.05.

injection of 15 mg/kg PA41, however, and found that the animals had a more severe outcome to a vegetative bacterial IP infection (S3 Fig). This presumably is due to an immune reaction to a high amount of foreign material. Instead, to have a higher amount of antibody circulating in the bloodstream, we performed sequential antibody dosing on days -5, -3, and -1 of 7.5 mg/kg to allow time for the animals to acclimate to the antibody administrations. Then, on day 0, animals were intoxicated with 10 or 50 ng TcsL and weighed and monitored for seven days (Fig 5A). PA41, and not CDB1, was able to neutralize the cytotoxic activity of 10 ng TcsL (Fig 5B). PA41 also showed efficacy in neutralizing up to 50 ng TcsL, and all animals survived the study (Fig 5C). Uterine tissues were harvested upon euthanization and processed for histology. H&E-stained tissue (S4 Fig) was scored from mild to severe in edema, acute inflammation, and epithelial injury by a pathologist blinded to the experimental conditions (Fig 5D). Scores of moderate to severe were assigned to animals that had been transcervically instilled with 50 ng TcsL. When PA41 was administered to pre-treat 50ng TcsL, scoring was reduced in all criteria. CDB1 administration did not improve the scoring in mice that had been treated with 10ng TcsL. Complete blood counts were performed on blood at time of death or at end of study. Total white blood cell (WBC) counts for 10 and 50ng TcsL showed no statistically significant difference compared to PA41, CDB1, and PBS treated mice (Fig 5E). However, in a WBC differential analysis, 50 ng TcsL alone showed an increase in neutrophils (NE) and a decrease in lymphocytes (LY) when compared to PBS control animals (Fig 5F). Additionally, hematocrit (HCT) levels, i.e., the proportion of red blood cells in the blood, was increased in animals instilled with 50 ng TcsL. Animals administered PA41/TcsL (50 ng) had NE, LY and HCT levels similar to PBS control mice. WBC differential analysis of animals treated with 10ng TcsL in the presence or absence of CDB1 were found to be similar to PBS control mice (Fig 5G).

## Prophylactic administration of PA41 in treatment of transcervical *P. sordellii* infection

All animals were administered medroxyprogesterone acetate five days prior to instillation to induce diestrus. Animals were intraperitoneally injected with 7.5 mg/kg PA41 or PBS one day prior to TC infection of $10^5$ CFUs vegetative *P. sordellii* (Fig 6A). Although not statistically significant, we did see a delay in mouse death at 36h post infection with 80% survival of animals treated with PA41 compared to 40% survival of PBS-treated mice. At the end of study at 96h, however, there was only a slight non-significant difference between PA41- and PBS- treated animals, with 40% and 30% overall survival, respectively (Fig 6B).

## PA41 in treatment of transcervical *P. sordellii* spore infection

In addition to evaluating vegetative bacteria, we wanted to test if the TC instillation model would be responsive to *P. sordellii* spores. Indeed, TC inoculation of $10^5$ and $10^6$ *P. sordellii* 9714 CFU spores in diestrus mice was found to be lethal, with $10^5$ CFU spores having a delayed onset of disease and animal mortality beginning after Day 6 (Fig 6C).

Finally, we wanted to assess if PA41 could be used in the treatment of TC *P. sordellii* 9714 spore infections. To test this, all animals were induced into diestrus five days prior to transcervical inoculation of $10^6$ spores. One-, three-, and five-days following infection, animals were intraperitoneally injected with 7.5 mg/kg PA41 or PBS. Animals were weighed daily and monitored for 7–10 days (Fig 6D). We found that animals administered PA41 following PSI had higher survival rates compared to PBS treated mice (Fig 6E). The differences fell short of statistical significance when using the log-rank (Mantel-Cox) multiple comparison test (p = 0.06) but were significant when using the Gehan-Breslow-Wilcoxon test that gives more weight for

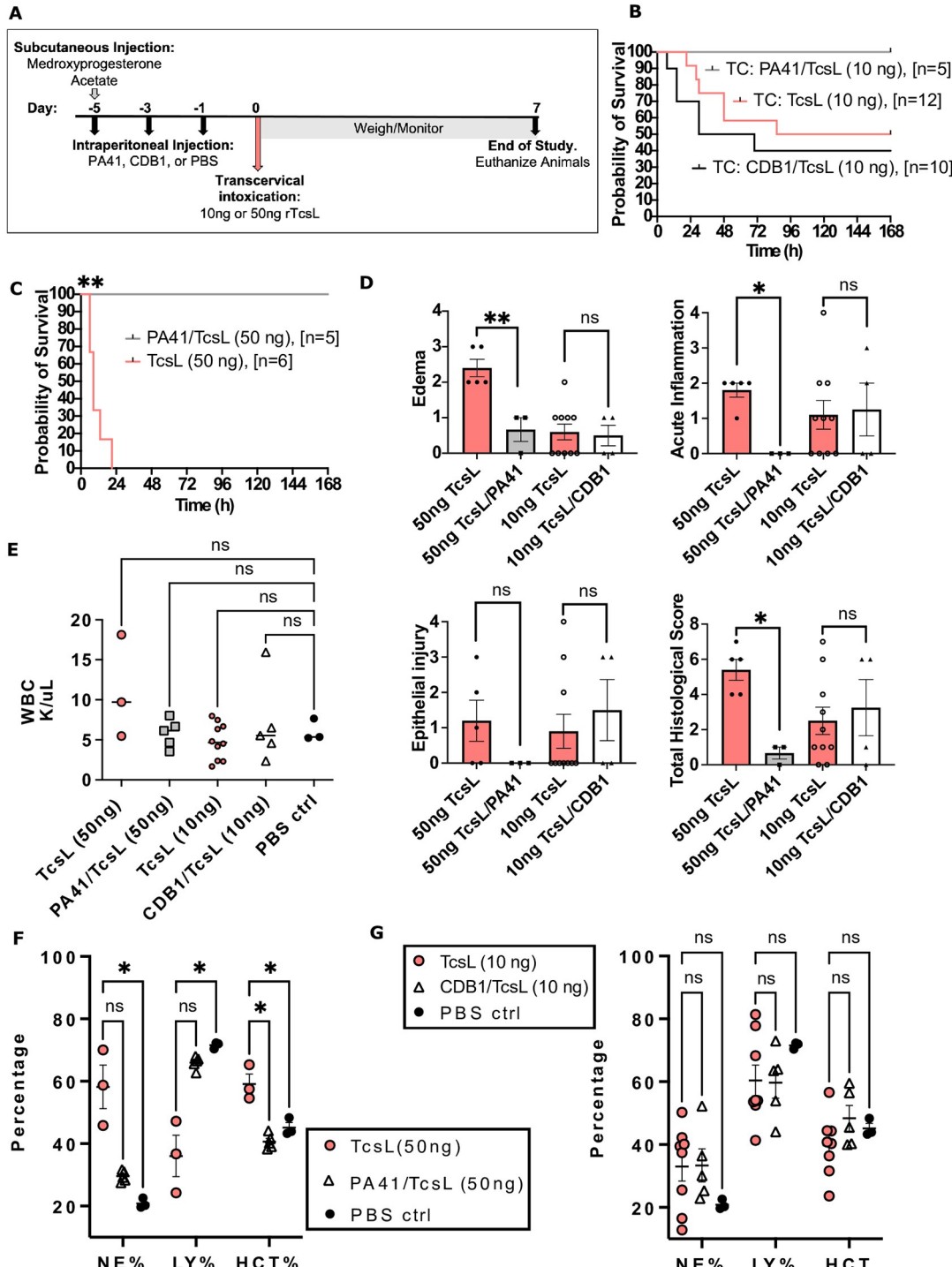

**Fig 5. Neutralization studies of monoclonal antibodies, PA41 and CDB1, following transcervical intoxication of rTcsL.**
**(A)** Timeline of neutralization studies: subcutaneous administration of medroxyprogesterone acetate on Day -5 to induce diestrus. Intraperitoneal injections of 7.5 mg/kg PA41, 7.5 mg/kg CDB1, or PBS on Days -5, -3, -1. Transcervical intoxication of rTcsL on Day 0. Animals weighed/monitored for seven days. **(B)** Survival curve of animals in diestrus treated with PA41 or CDB1 on days -5, -3, -1, and transcervically intoxicated with 10ng TcsL on Day 0. **(C)** Survival curve of animals in diestrus treated with PA41, on days -5, -3, -1, and transcervically intoxicated with 50ng TcsL on Day 0. Log-rank (Mantel-Cox) multiple comparison test was used with statistical significance set at a p value of <0.05. **(D)** Histological scoring of edema, acute inflammation, and epithelial injury of uterine tissues at time of death or end of study from mice transcervically instilled with 50ng TcsL in the presence or absence of PA41 and 10ng TcsL in the presence of absence of CDB1. Mann-Whitney test was used

with statistical significance set at a p value of <0.05. **(E)** White blood cells count following Complete Blood Counts. Neutrophil (NE), lymphocyte (LY) and hematocrit (HCT) blood cell percentages of **(F)** TcsL/PA41 and **(G)** TcsL/CDB1. Kruskal-Wallis multiple comparison test was used with statistical significance set at a p value of <0.05.

earlier timepoints (0.04). These data support our overall conclusion that PA41 can reduce the impact of PSI in a mouse model of uterine infection.

## Discussion

Reproductive-age women are at increased risk for PSI because this organism can cause a nearly 100% fatal intrauterine infection following childbirth or abortion [1]. When women present with PSI-associated TSS, there is very little information on how to treat the patient [1]. Antibiotics can be used to treat the *P. sordellii*, but the toxins remain active. Antitoxin preparations against lethal toxin were shown to prevent cytotoxicity of culture supernatants of *P. sordellii*, as well as *C. difficile*, in cell culture and lethality in mouse studies [8]. However, there is no commercially available antitoxin for treatment of human infection. A therapeutic drug that targets TcsL, the key virulence factor in PSI, could significantly reduce mortality in these patients.

In this study, we tested *C. difficile* anti-TcdB mABs, PA41 and CDB1, for their capacity to provide protection against TcsL. TcsL and TcdB share, not only a high level of sequence identity (~ 76% identity) and structural homology, but also similar mechanisms of intoxication. Antibody cross-reactivity was reported when TcsL was first purified and characterized, showing an antibody that neutralized TcsL was also able to recognize and bind TcdB [8]. This report is consistent with our finding that both anti-TcdB mABs significantly protected Vero cells from the cytotoxic activity of TcsL, though PA41 appeared to have better efficacy (Fig 1).

PA41 and Bezlotoxumab (which shares epitope binding sequences with CDB1) interactions with TcdB have been characterized and their epitopes and modes of neutralization are known [4–6]. PA41 binds the GTD and prevents the translocation of the enzymatic domain into the host cell [4]. Bezlotoxumab, on the other hand, functions to block TcdB binding to the CSPG4 host cell receptor [6,12]. Due to their high sequence identity, it is presumed that TcdB and TcsL have similar antibody epitopes. For example, the high sequence identity between TcdB and TcsL at the known TcdB/PA41 interface suggests a similar mechanism of antibody neutralization (S5 Fig). It is likely, that in the presence of PA41, TcsL is able to bind and enter the host cell, but the enzymatic domain is unable to be translocated into the cytosol and thereby unable to inactivate host GTPases. In the case of CDB1, it is perhaps unsurprising that this mAb was less effective than PA41 in TcsL neutralization, as the TcsL receptors in the Sema6 family bind the TcsL delivery domain, in a distinct location from the CSPG4 and Bezlotoxumab binding sites [13,14].

Nevertheless, we wanted to test both mAbs for their neutralization efficacy against TcsL *in vivo*. Using a murine IP intoxication model, we observed that both PA41 and CDB1 were able to neutralize a lethal IP dose of TcsL (2.5 ng) and protect the animals from death (Fig 2B). Similarly, PA41 increased mouse survival following IP injection of vegetative *P. sordellii* bacteria (Fig 2D).

To test PA41 and CDB1 neutralization of TcsL and *P. sordellii* in a more physiologically relevant animal model, we developed a transcervical inoculation method. The method allows for a non-surgical transfer of inoculum through the vaginal orifice, past the cervix, and directly into the uterus. By eliminating the surgical laparotomy method used in a prior model [10], we expected to minimize the risk of introducing an undesired infection and cut out a surgical recovery period. However, initially we found that TcsL was not able to cause disease when

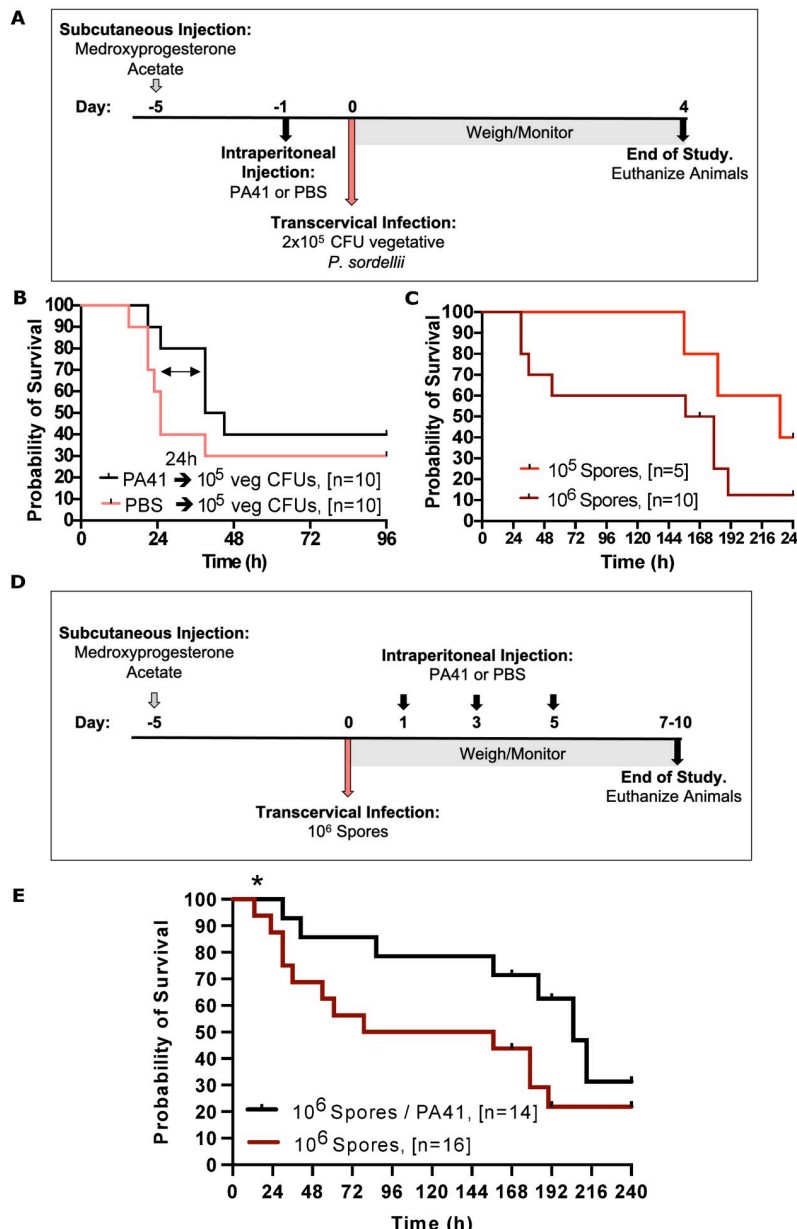

**Fig 6. Prophylactic and therapeutic administration of PA41 following transcervical *P. sordellii* vegetative bacterial or spore infection, respectively. (A)** Timeline showing subcutaneous administration of medroxyprogesterone acetate on Day -5 to induce diestrus, intraperitoneal injection of 7.5 mg/kg PA41 or PBS on Day -1, and transcervical infection of $10^6$ vegetative *P. sordellii* bacteria on Day 0. Animals were weighed/monitored for four days. **(B)** Survival curve of diestrus animals prophylactically treated with PA41 or PBS and TC inoculated with $2 \times 10^5$ CFUs vegetative bacteria. **(C)** Survival curve following transcervical inoculation of $10^5$ and $10^6$ spores in mice in diestrus. **(D)** Timeline showing subcutaneous administration of medroxyprogesterone acetate on Day -5 to induce diestrus, transcervical infection of $10^6$ spores on Day 0 and intraperitoneal injection of 7.5 mg/kg of PA41 or PBS one-, three-, and five-days post-infection. Animals were weighed/monitored for 7–10 days. **(E)** Survival curve of diestrus animals treated with PA41 or PBS following TC inoculation of $10^6$ spores. Gehan-Breslow-Wilcoxon test was used with statistical significance set at a p value of <0.05. *Gehan-Breslow-Wilcoxon p = 0.0424, Log-rank p = 0.0624.*

given transcervically (Fig 3B), and vegetative *P. sordellii* infection resulted in minimal, inconsistent disease in the mice (Fig 3C).

We tested if female sex hormones play a role in the pathogenesis of infection. We used estrogen and progesterone to induce prolonged stages of estrus (ovulation) and diestrus (sexual quiescence), respectively. Animals in diestrus were found to have a more adverse outcome following transcervical TcsL intoxications (Fig 4B) and *P. sordellii* infections (Fig 4D) compared to animals in estrus. Our data complement findings of other investigators who have reported that uterine ascending infections of *Neisseria gonorrhoeae*, *Chlamydia trachomatis*, and Herpes simplex virus type-2 in mice show profoundly different disease outcomes at different stages of the reproductive cycle [15–17]. For example, in the case of *Neisseria gonorrhoeae*, under the influence of progesterone, significant epithelial remodeling allows gonococcal entry into the underlying stroma [15]. We speculate that the epithelial remodeling associated with diestrus is allowing TcsL to access endothelial cells and the blood stream. In addition, it is known that in estrus there is an increased production of mucus in the uterus. Presumably, this could give the animals a layer of protection preventing toxin from reaching the epithelium of the uterus. It is also possible that other factors, such as *P sordellii* toxin receptors may be differentially expressed under differing hormone treatments. Additional studies are needed to understand how different hormonal environments impact *P. sordellii* pathogenesis in the genital tract.

Having established a hormone-dependent transcervical inoculation method, we again tested PA41 and CDB1 for their capacity to neutralize TcsL toxicity. In this model, we found that PA41 was able to protect mice from TcsL lethality but CDB1 was not (Fig 5B and 5C). Although it would have been exciting to see protection in CDB1-treated mice, given the clinical availability of the related Zinplava mAB, it is reasonable that the differences between TcsL and TcdB receptor specificity account for this lack of efficacy.

We further analyzed the uterine tissues from PA41-treated mice that were intoxicated with TcsL and found reduced levels of edema, inflammation, and epithelial damage when compared to PBS-treated TcsL intoxicated mice (Fig 5D). A characteristic of PSI is the onset of a leukemoid reaction (LR), i.e., a significant increase in white blood cells. We did not observe a significant difference in white blood cell numbers between TcsL and PBS instilled mice, suggesting that TcsL alone is not responsible for the LR (Fig 5E). In TcsL instilled mice, compared to PBS-control mice, we did, however, observe a shift in WBC differential counts where lymphocytes (LY) were decreased, and neutrophils (NE) were increased (Fig 5F). We also observed an accumulation of fluid in the thoracic cavity of these mice suggesting an increased permeability of the vascular system. This increased permeability could account for the increased hematocrit (HCT) found in animals instilled with TcsL, where the blood becomes concentrated with RBCs (Fig 5F). Animals administered PA41/TcsL had NE, LY and HCT levels similar to those of PBS control mice (Fig 5F).

We were curious to move forward with PA41 to determine its efficacy against vegetative *P. sordellii*. We began with prophylactic administration 24hr prior to transcervical infection. Although the resulting survival curve was not statistically significant, there does appear to be a delay in mortality in animals treated with PA41 compared to PBS at approximately 36h post infection (Fig 6B). At this timepoint, PA41 neutralization appeared to be occurring but this was rapidly followed by survival decline. With only a single mAb administration, it is possible that PA41 is capable of neutralization at a 36h timepoint but is being depleted and can't keep up with additional bacterial production of TcsL. Perhaps additional administrations of PA41 and/ or incorporation of antibiotic therapy to deplete bacterial reproduction would demonstrate further efficacy of PA41. In addition, the organism produces several additional virulence

factors, e.g., a sialidase and phospholipase C that play unknown roles in PSI. These are ideas we plan to explore in future studies.

Lastly, we show that *P. sordellii* spores can germinate and cause disease when given trans-cervically to mice in diestrus (Fig 6C) and that PA41 treatment can lead to increased survival relative to PBS-treated mice (Fig 6E). There are several variables and questions for follow-up study. For example, what are the germinant and environmental conditions within the host that affect the efficiency of spore germination and is this influenced by the reproductive cycle of the mice? Would the level of protection improve if using a mouse-derived mAB or the addition of antibiotics? We hope that the availability of this relatively easy uterine infection model will provide a system to address these fundamental questions and facilitate the work needed to advance candidate therapeutics for addressing human PSI.

## Materials and methods

### Ethics statement

This study was approved by the Institutional Animal Care and Use Committee at Vanderbilt University Medical Center (VUMC) and performed using protocol M1700185-01. Our laboratory animal facility is AAALAC-accredited and adheres to guidelines described in the Guide for the Care and Use of Laboratory Animals. The health of the mice was monitored daily, and severely moribund animals were humanely euthanized by $CO_2$ inhalation.

### Recombinant *P. sordellii* toxin purification

TcsL was amplified from *P. sordellii* strain JGS6382 and inserted into a BMEG20 vector (Mobi-Tec) using BsrGI/KpnI restriction digestion sites in the vector, as reported previously [18]. Plasmids encoding His-tagged TcsL (pBL552) were transformed into *Bacillus megaterium* according to the manufacturer's protocol (MoBiTec). Six liters of LB medium supplemented with 10 mg/liter tetracycline was inoculated with an overnight culture to an optical density at 600 nm (OD600) of $\sim 0.1$. Cells were grown at 37°C and 220 rpm. Expression was induced with 5 g/liter of d-xylose once cells reached an OD600 of 0.3 to 0.5. After 4 h, the cells were centrifuged and resuspended in 20 mM HEPES (pH 8.0), 500 mM NaCl, and protease inhibitors. An EmulsiFlex C3 microfluidizer (Avestin) was used to generate lysates. Lysates were then centrifuged at $40,000 \times g$ for 20 min. Supernatant containing toxin was run over a Ni-affinity column. Further purification was performed using anion-exchange chromatography (HiTrap Q HP, GE Healthcare) and gel filtration chromatography in 20 mM HEPES (pH 6.9), 50 mM NaCl.

### Monoclonal antibodies

PA41 and PA50 were supplied by AstraZeneca (previously MedImmune). CDB1 DNA constructs for the light chain and heavy chain equivalents of Bezlotoxumab were synthesized and cloned into custom plasmids encoding the heavy chain IgG1 constant region and the corresponding kappa light chain region (pTwist 314 CMV BetaGlobin WPRE Neo vector, Twist Bioscience). The antibodies were transiently expressed in Expi-293F mammalian cells with PEI transfection reagent. Cells were cultured in FreeStyle F17 expression Medium supplemented with 10% Pluronic F-68 and 10% GlutaMAX until expression was terminated 5–7 days post-transfection. The mAb was isolated by protein A affinity (HiTrap Protein A HP, 17-0403-01, GE Healthcare) according to the manufacturer's instructions. All mAb administrations were given via IP.

### Vero cell culture and viability assays

Vero cells were maintained in DMEM supplemented with 10% fetal bovine serum and cultured at 37˚C with 5% $CO_2$. Cells were seeded into 96-well plates at 1,500 cells per well and allowed to grow overnight. For intoxication, toxin and mABs were diluted in DMEM/FBS and incubated together for 1hr at 37˚C. Toxin/mAB mix was incubated on the cells for 72hr at the concentrations indicated and viability was measured using the CellTiter-Glo luminescent cell viability assay (catalog number G7573; Promega). Dose response curves were plotted and fit to a sigmoidal function (variable slope) to determine $EC_{50}$ using Prism software (GraphPad Prism Software).

### *P. sordellii* vegetative and spore preparation

*P. sordellii* strain ATCC 9714 was obtained from David Aronoff and cultured at 37˚C in an anaerobic chamber (90% nitrogen, 5% hydrogen, 5% carbon dioxide). For vegetative bacteria, a single colony was picked to inoculate Reinforce Clostridial Medium (RCM) [BD, 21081], followed by incubation overnight. A 10 mL RCM subculture (OD600 = 0.05) was prepared and allowed to grow for 2-3h. The OD600 was measured and CFUs were determined from a previous growth curve. The culture was centrifuged and washed three times with PBS to remove any secreted toxins. The bacterial pellet was resuspended in desired CFUs/mL. For spore preparation, a single colony was picked to inoculate 10 mL RCM culture, followed by incubation overnight at 37˚C. The next day, 2 mL of that culture was inoculated into 2 mL Columbia broth for overnight growth at 37˚C. The next day, 4mL of that culture was inoculated into 40 mL of Clospore medium, followed by growth for 7 days. The culture was centrifuged and washed three times in cold sterile water. Spores were suspended in 1 ml of sterile water and heat treated at 65˚C for 20 min to eliminate vegetative cells. Viable spores were enumerated by CFU on RCM plates. Spore stocks were stored at 4˚C until use.

### Animals and housing

All mouse experiments were approved by the Vanderbilt Institutional Animal Care and Use Committee (IACUC). C57BL/6J mice (all females, age 9 to 12 weeks) were purchased from Jackson Laboratories and were housed five to a cage in a pathogen-free room with clean bedding and free access to food and water. Mice had 12h cycles of light and dark.

### Virulence studies

For intraperitoneal intoxications and infections, mice were anesthetized and intraperitoneally injected with recombinantly purified TcsL or vegetative *P. sordellii* bacteria alone or in the presence of mAb. For transcervical instillation, mice were anesthetized, and a speculum was inserted into the vaginal cavity to allow for dilation and passage of a flexible gel-loading pipette tip through the cervix and transfer of recombinant protein, vegetative bacteria, or spores directly into the uterus. Following instillation, a cotton plug applicator was inserted into the vagina, and a cotton plug was expelled from the applicator and into the vaginal cavity using a blunt needle. Mice were monitored daily for morbidity and signs of sickness. Mice were humanely euthanized by $CO_2$ inhalation when moribund or at end of study. In some cases, the uterus was harvested, fixed, paraffin-embedded, and processed for histology.

### Hormone administration and estrous cycle staging

Mice were subcutaneously injected with water soluble beta-estradiol (0.5 mg/mouse, Sigma Aldrich) two days prior to infection to prolong estrus, or medroxyprogesterone acetate (2 mg/

mouse, Amphastar Pharmaceuticals) five days prior to infection to synchronize in diestrus. Immediately prior to infection, the estrous stages of the animals were confirmed via vaginal lavage. To accomplish this, the vagina was washed with 20uL saline using a 20 μl micropipette. Wet smears were examined under 40x objective and the stage of the estrous cycle determined based on cytology [19].

## Statistical analysis

Statistical testing and graphical representations of the data were performed using Graphpad Prism (Statistical significance was set at a $P \leq 0.05$ for all analyses (*, $P \leq 0.05$; **, $P \leq 0.01$; ***, $P \leq 0.001$; ****, $P \leq 0.0001$). The Log-rank (Mantel-Cox) multiple comparison test was used for survival curve comparisons. The Gehan-Breslow-Wilcoxon test that gives more weight for earlier timepoints, was used for Fig 6E survival curve comparison. The Mann-Whitney-Wilcoxon rank sum (Mann-Whitney) test was used to compare two groups, or the Kruskal-Wallis test was used to calculate significance using Dunn's test when two groups were compared within multiple comparisons.

## Supporting information

**S1 Fig. Mice intoxicated with TcsL had a buildup of fluid in the thoracic and peritoneal cavities.** Pleural effusion collected at time of euthanasia of two moribund mice following IP intoxication of 2.5 ng TcsL. Following PA41 administration, mice were protected from TcsL-induced pleural effusion.
(TIF)

**S2 Fig. PA41 is present in blood serum of animals following intraperitoneal administration.** Western blot analysis of PA41 in mouse serum using an anti-human Fab antibody on 1, 2, and 3 days post IP injection of 7.5 mg/kg mAb. Control PA41 was included on the blot for comparison.
(TIF)

**S3 Fig. Dosing with higher concentrations of PA41 does not improve the survival outcome for mice infected IP with vegetative P. sordellii.** Mouse survival curve following IP infection of vegetative *P. sordellii* bacteria ($<10^7$ CFUs) alone or in the presence of 15 mg/kg PA41.
(TIF)

**S4 Fig. H&E staining for uterine histology following transcervical instillation. (A)** Schematic depicting the mouse uterus. 10x brightfield images of H&E staining of mouse uterine horn tissue following transcervical instillation of PBS **(B)**, 50 ng TcsL **(C)**, 50 ng TcsL / PA41 **(D)**, 10 ng TcsL **(E)**, or 10 ng TcsL / CDB1 **(F)**.
(TIF)

**S5 Fig. The high sequence identity between TcdB and TcsL at the known TcdB/PA41 interface suggest a similar mechanism of antibody neutralization.** Sequence alignment of TcdB-GTD and TcsL-GTD subdomains containing the TcdB/PA41 epitope. Conserved residues are highlighted in red, with PA41 epitope residues denoted by asterisks.
(TIF)

## Acknowledgments

We gratefully acknowledge Paul Warrener at AstraZeneca who provided the PA41 mAB used in this study and the Vanderbilt University Medical Center Translational Pathology Shared

Resource for assistance with tissue embedding and blood analyses. The Translational Pathology Shared Resource is supported by NCI/NIH Cancer Center Support Grant P30CA068485.

## Author Contributions

**Conceptualization:** Sarah C. Bernard, D. Borden Lacy.

**Data curation:** Sarah C. Bernard.

**Formal analysis:** Sarah C. Bernard, M. Kay Washington, D. Borden Lacy.

**Funding acquisition:** D. Borden Lacy.

**Investigation:** Sarah C. Bernard.

**Methodology:** Sarah C. Bernard, D. Borden Lacy.

**Project administration:** D. Borden Lacy.

**Resources:** Sarah C. Bernard, D. Borden Lacy.

**Supervision:** D. Borden Lacy.

**Validation:** Sarah C. Bernard, D. Borden Lacy.

**Visualization:** Sarah C. Bernard.

**Writing – original draft:** Sarah C. Bernard.

**Writing – review & editing:** D. Borden Lacy.

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
