## [Decision Letter · Decision Letter 0]

16 Aug 2022

Dear Borden,

Thank you very much for submitting your manuscript "Mechanisms of virulence and protection in Paeniclostridium sordellii infection" for consideration at PLOS Pathogens. As with all papers reviewed by the journal, your manuscript was reviewed by members of the editorial board and by several independent reviewers. In light of the reviews (below this email), we would like to invite the resubmission of a significantly-revised version that takes into account the reviewers' comments.

As you will read, the overall assessments of the manuscript by the two referees were substantially different. This incongruence is at least partially due to differences in perceptions of the importance and impact of the manuscript's results. Despite enthusiasm expressed by Referee #1 (in particular), both referees communicated that substantial additional experimentation will be required to strengthen the overall conclusions and impact of the manuscript.

Regrettably, we cannot make any decision at this time about publication until we have seen the revised manuscript and your response to the reviewers' comments. Your revised manuscript is also likely to be sent to reviewers for further evaluation.

Sincerely,

Steven R. Blanke

Associate Editor

PLOS Pathogens

Karla Satchell

Section Editor

PLOS Pathogens

Kasturi Haldar

Editor-in-Chief

PLOS Pathogens

orcid.org/0000-0001-5065-158X

Michael Malim

Editor-in-Chief

PLOS Pathogens

orcid.org/0000-0002-7699-2064

Reviewer's Responses to Questions

**Part I - Summary**

Reviewer #1: In this manuscript, Bernard et al. report on an important study on the development of a robust physiologically-relevant animal model for a P.sordelli infection (PSI), and the efficacy of two monoclonal antibodies against TcsL, the primary virulence determinant of P.sordellii. Though PSI is a rare condition, it is almost universally fatal when contracted by women following childbirth, stillbirth or abortion and there are no approved therapeutics to address secreted TcsL. This is due, in part, to a lack of good animal models.

In this manuscript Dr. Lacy's group first investigated the efficacy of two monoclonal antibodies, PA41 and CDB1, that were originally developed against TcdB - a close homolog of TscL. They show in vitro that PA41, and to a lesser extent CDB1, prevent TcsL-ionduced cytotoxicity. PA41 and CDB1 protected mice against an IP injection of a lethal dose of TcsL, with PA41 showing superior protection. In the second part of the manuscript, the authors benchmark the TC instillation model of recombinant TcsL and vegetative P.sordellii and find, surprisingly, that there were no signs of disease or death. They go on to show that the murine reproductive cycle has a profound impact on the pathogenic effects of PSI; remarkably, where as estrus animals were completely resistant to disease induced by toxins, diestrus animals were completely sensitive. Similar effects were observed upon infection with vegetative P.sordellii in the TC model. Finally, they used this model for mice in diestrus to test the efficacy of pre-dosed PA41 and CBD1 against TcsL, and vegetative bacteria and saw as before superior efficacy for PA41. Finally, they demsontrate a protective effect of PA41 in a spore-challenge model.

Strengths: This is an important study that shows quite conclusively the effects of the reproductive cycle on PSI disease pathogenesis. The breadth of conditions explored in Figures 4 and 5 is truly impressive and demonstrates the robustness of the model. This work also opens up new avenues for exploring PSI pathogenesis and how epithelial remodelling associated with diestrus impacts disease. Furthermore, this work shows for the first time that a monoclonal antibody against TcsL (PA41) may be a viable approach to treat PSI and disease.

Weaknesses: While the experiments are generally well done, there are some additional experiments requested (see below) that are needed in my view to strengthen the work, and round out the data.

Reviewer #2: In this paper, the authors evaluated the neutralizing effects of two TcdB antibodies on TcsL. The authors found that monoclonal antibody PA41 can neutralize TcsL both in the cell and mouse model, which might be developed as a potential avenue for treating P. sordellii. However, given the high sequence similarity between TcsL and TcdB, it is not surprising that some TcdB antibodies can also neutralize TcsL. Indeed, cross reactivity of antiserum against TcdB and TcsL has been earlier reported. Thus, the current study has limited scientific and clinical significance in understanding the P. sordellii pathogenesis or developing novel therapeutic avenues.

**Part II – Major Issues: Key Experiments Required for Acceptance**

Reviewer #1: Considering how straightforward the in vitro experiments are to perform, as compared with the in vivo experiments, it is surprising that the authors did not provide a more complete set of experiments in Figure 1. In particular, the authors should include a dose titration of both antibodies (at a cytotoxic dose of TcsL) to get a sense of their relative potencies. Furthermore, it would be desirable if the data of the titration of toxin in the presence and absence of a fixed dose of antibody were presented more clearly. A full titration of toxin, perhaps with small increments shown as curves plus and minus antibody would be preferable. The inclusion of red bar for No AB is confusing also.

[optional] In Figure 5, rather than pre-dose animals with antibodies before the toxin challenge, did the authors consider dosing the antibodies right before or right after the TC challenges were performed. Without any knowledge of the PK of the antibodies, it is difficult to know how much antibody is circulating. The efficacy might be better if dosed differently.

The marginal efficacy seen in Figure 6E in "treatment mode" was underwhelming. Did the authors consider dosing higher? There was no specific rationale given for dosing at 7.5mpk. If it is not dose-limiting, it would be important to explore whether a higher dose of antibody might work in this paradigm.

Reviewer #2: 1. There are many antibodies/nanobodies/immune molecules developed that can effectively neutralize TcdB. But only PA41 and CDB1 were tested. The authors should give a more comprehensive evaluation of other known TcdB antibodies/nanobodies/immune molecules and compare their neutralizing effects against TcsL.

2. The neutralizing effects of PA41 and CDB1 are not characterized quantitively. What are the IC50 values in the cell models. Also, multiple cell lines need to be tested. What are the binding affinities (KD) of these antibodies to TcsL, and compared to TcdB?

3. P. sordellii produces two major exotoxins, TcsL and TcsH. Although less toxic, TcsH causes strong hemorrhagic effects and should not be ignored. A combination of neutralizing agents against both TcsL and TcsH may bring optimized protection against P. sordellii. In particular, a recent study has defined TMPRSS2 as a receptor for TcsH, which can help to design neutralizing molecules against TcsH.

4. In the previous studies, lung was proposed as a vulnerable target for TcsL. Why this lung damage model was not used to evaluate the protection effect of PA41? Also, studies showed that soluble SEMA6A/B can effectively protected mouse from TcsL-induced lung damage. What is the neutralizing efficacy of PA41 compared to SEMA6A/B in vivo?

5. In transcervical infection models, the difference of TcsL/P. sordellii susceptibility between mice in diestrus and estrus may simply be explained by morphological and physiological changes of cervix. For example, minor wounds and increased permeability of the epithelium during estrus. To demonstrate this (or not), more pathological and biochemical analysis on cervix need to be performed. Also, IP injection of TcsL/P. sordellii in mice at diestrus and estrus state should be performed as a comparison.

**Part III – Minor Issues: Editorial and Data Presentation Modifications**

Reviewer #1: - The title of the paper is too vague. It sounds like a review paper title. Please fix this to make it more informative.

- The authors should provide some rationale for the doses of antibodies used throughout this study (0.75mpk ->7.5mpk)

- The data in Figure 3D are marginal. The language on Line 141-142 could be softened.

Reviewer #2: 1. The mechanism of virulence is merely mentioned, and the title is overstated.

2. The numbers of mice used in some groups were not sufficient to obtain reliable conclusions, such as TcsL/CDB1 in Fig2B and IP-106 in Fig2C.

3. Since co-structure of PA41-TcdB is known (actually from the same group), it would be ideal to compare the consensual and divergent residues at the interface between PA41-TcdB and PA41-TcsL.

4. Fig 3. How did the author control the estrous cycle stage of the mice here?

PLOS authors have the option to publish the peer review history of their article (what does this mean?). If published, this will include your full peer review and any attached files.

Reviewer #1: No

Reviewer #2: No
---

## [Editor Report · Decision Letter 1]

10 Nov 2022

Dear Dr. Lacy,

We are pleased to inform you that your manuscript 'Paeniclostridium sordellii uterine infection is dependent on the estrous cycle' has been provisionally accepted for publication in PLOS Pathogens.

Best regards,

Steven R. Blanke

Academic Editor

PLOS Pathogens

Karla Satchell

Section Editor

PLOS Pathogens

Kasturi Haldar

Editor-in-Chief

PLOS Pathogens

orcid.org/0000-0001-5065-158X

Michael Malim

Editor-in-Chief

PLOS Pathogens

orcid.org/0000-0002-7699-2064
---

## [Editor Report · Acceptance letter]

17 Nov 2022

Dear Dr. Lacy,

We are delighted to inform you that your manuscript, "Paeniclostridium sordellii uterine infection is dependent on the estrous cycle," has been formally accepted for publication in PLOS Pathogens.

Best regards,

Kasturi Haldar

Editor-in-Chief

PLOS Pathogens

orcid.org/0000-0001-5065-158X

Michael Malim

Editor-in-Chief

PLOS Pathogens

orcid.org/0000-0002-7699-2064